# The combined effects of temperature and relative humidity parameters on the reproduction of *Stomoxys* species in a laboratory setting

Arman Issimov[1]*, David B. Taylor[2], Kuandyk Zhugunissov[3], Lespek Kutumbetov[3], Assylbek Zhanabayev[4], Nurlybay Kazhgaliyev[4], Aliya Akhmetaliyeva[5], Birzhan Nurgaliyev[5], Malik Shalmenov[5], Gaisa Absatirov[5], Laura Dushayeva[5], Peter J. White[1]

1 Sydney School of Veterinary Science, Faculty of Science, University of Sydney, Sydney, Australia, 2 Agroecosystems Management Research Unit, USDA-ARS, Lincoln, NE, United States of America, 3 RGE "Research Institute for Biological Safety Problems" Committee of Science, The Ministry of Education and Science of the Republic of Kazakhstan, Nur-Sultan, Kazakhstan, 4 Department of Veterinary Medicine, Saken Seifullin Kazakh Agrotechnical University, Nur-Sultan, Kazakhstan, 5 Department of Veterinary Medicine, Zhangir Khan West Kazakhstan Agrarian–Technical University, Uralsk, Kazakhstan

* issimovarman@gmail.com, aiss0820@uni.sydney.edu.au

## Abstract

In this study, *Stomoxys* species (*S. calcitrans*, *S. sitiens and S. indica*) were examined to improve on the current technique for mass rearing using a method of combined incubation parameters. Moreover, the reproductive potential of immature forms at various stages of development was defined. Immature forms of stable flies were incubated according to species. There was no significant difference in the number of immature forms obtained among species incubated under the same conditions. Six incubation parameters were used in combination, at temperatures (T) of 32˚C, 27˚C and 22˚C and relative humidity (RH) of 90% and 70% RH. The combined method resulted in a higher number of eggs hatching at 32˚C and 90% humidity as well as an increase in the number of larva pupated and emergence of imago at 27˚C and 70% humidity.

## Introduction

Stable flies (*Stomoxys* spp) have long been considered a major pest of livestock in Kazakhstan and are capable of transmitting pathogens present in infected animals [1]. A high concentration of these flies is observed in almost all regions in Kazakhstan with adult *Stomoxys* flies emerging in huge numbers following rainfall events in the later months of spring. Farms are a preferred habitat for stable flies where they can be found in large numbers in association with livestock [2, 3]. They interrupt grazing of cattle and cause painful bites, which in turn leads to weight loss in adult and yearling cattle as well as reduced milk yield [4–6]. As a result, farmers have developed strategies to minimize direct contact between animals and biting flies, through

**Funding:** Issimov, A. This study was funded by research project # AP05135323 of the Science Committee/ Ministry of Education and Science/ Republic of Kazakhstan.

**Competing interests:** The authors have declared that no competing interests exist.

pasturing animals in the early morning and in the evening while keeping them in enclosed barns during the daytime when the biting insect numbers are highest.

Stable flies are mechanical vectors for many infectious pathogens [7]. In previous work, we defined the role of *Stomoxys* species in the transmission of Lumpy Skin Disease Virus (LSDV) [8]. To do so, a constant supply of large numbers of *Stomoxys* flies reared in laboratory conditions was required. Attempts to produce flies using previously published methods [9, 10] were unsuccessful in obtaining a high yield and as result the aim of this study was to improve the reproduction of three species of *Stomoxys* using various ambient temperature and relative humidity settings on the different developmental stages of flies.

## Materials and methods

### Fly catching

Two local farms in the vicinity of the Research Institute for Biological Safety Problems (RIBSP) in Zhambyl oblast, Kazakhstan practising mixed dairy and beef management were examined for the presence of a population of stable fly. Three *Stomoxys* species were identified, namely *S. calcitrans*, *S. sitiens* and *S. indica*, with *S. sitiens* considered dominant over the other species.

Farms mentioned above are on the RIBSP balance and no approval was required to access the collection sites.

A fly-catching unit (pyramid) consisting of a cellophane-coated three-piece pyramid-shaped structure was used to collect flies (Fig 1A). The width of the construction at its base was 100 cm and 120 cm height. At the tip of the pyramid, there was a hole of 10 cm in diameter, on which a container was installed to collect flies that enter it as they fly up along the inside of the pyramid. The unit was installed directly onto a dung heap so that the distance between the lower part of the pyramid and the dung was 5–7 cm. Similarly, several units were placed along haystacks. In addition to the unit described above, a commercial fly catching unit "Miniature CDC light trap with UV light" (USA) (Fig 1B) was used. The light trap was suspended from the ceiling of cattle barns. The fly traps were monitored every two hours for the presence of *Stomoxys spp*. Each batch of newly caught *Stomoxys spp* was transported immediately to the fly rearing compartment of the Entomology section of the RIBSP for further experimentation. Approximately 780 field collected flies were kept for rearing in glass cages.

### Morphological identification

Morphological identification was carried out using Xper$^2$ software (LIS, France) and the protocol based on Zumpt [11]. Stable flies were anaesthetized using $CO2$ in a gas chamber and placed onto the light board with a Bi-Aspheric magnification lens (COIL 4212 Fixed Stand Magnifier - 12X). Morphological characteristics of *Stomoxys* spp, for example, venation of wings, dorso-abdominal feature, legs and thoracic pattern were defined utilizing a stereomicroscope as well as a table with the morphological features for each species [12]. *Stomoxys spp* were then sorted according to species and allowed to recover. The mean recovery time was established at 20 min post-anaesthesia. The mortality rate due to $CO2$ exposure constituted up to 1% per batch. Each batch contained 60 flies of mixed *Stomoxys spp*.

### Fly rearing

**Preparation of cages for oviposition.** A method of rearing stable flies described by Parr (1959) was modified to obtain sufficient egg production and increased larval survival. The main reason for the modification was the extremely low hatching and larvae survival rates during the first instar obtained from the previous method.

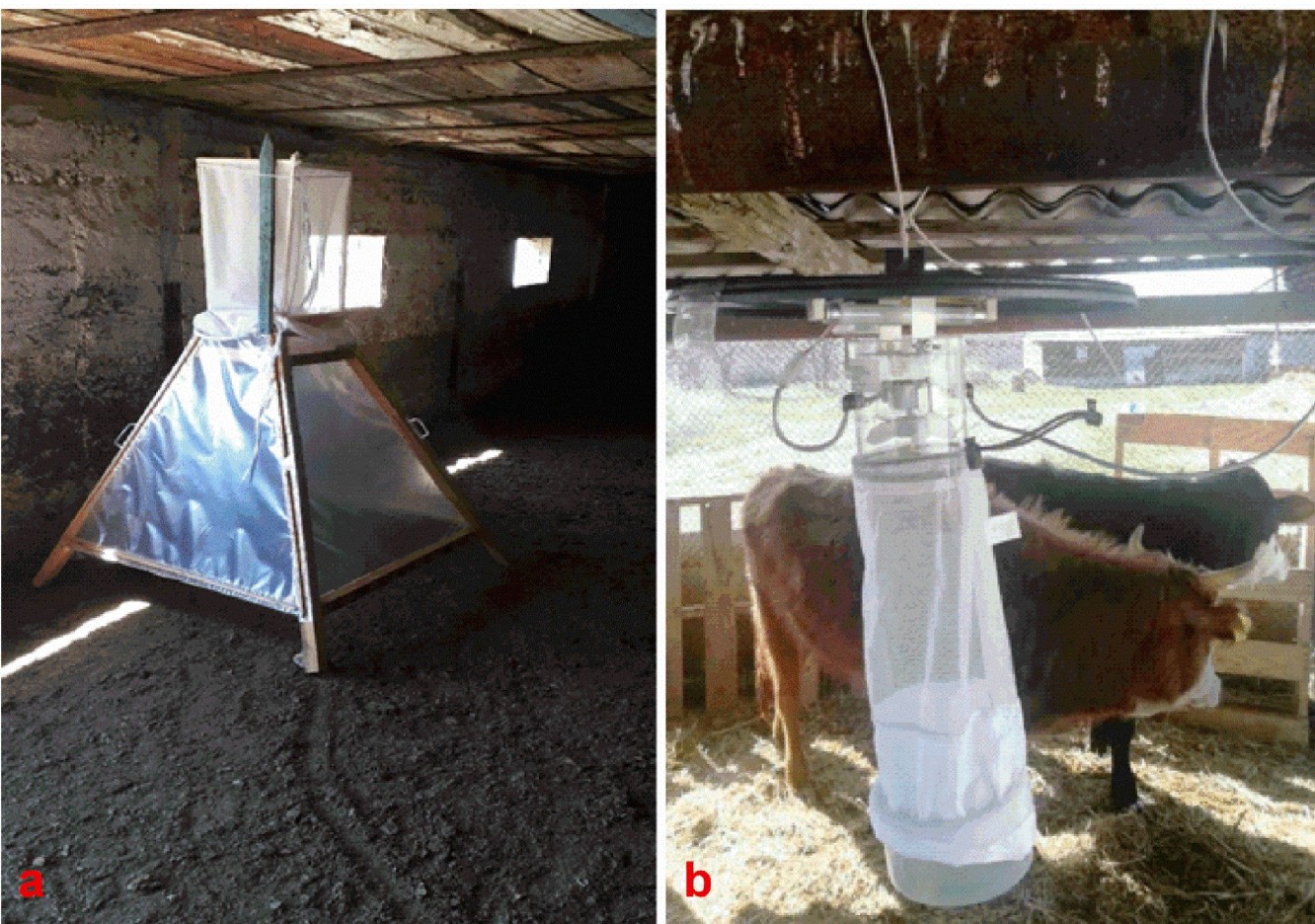

**Fig 1.** a–fly-catching unit (pyramid). b–Miniature CDC light trap with UV light.

The walls of the quadrangular cages for oviposition measured 45cm x 45cm x 1000cm and were built from wooden sheets (floor, back and sidewalls) and transparent plastic (Plexiglas) (top and front walls). This construction provided excellent illumination that directly affected the activity of the *Stomoxys* flies. On the top of the cage there were four holes cut 4 cm in diameter each fitted with removable mosquito net. They were located 20 cm from each other to allow air circulation and these holes were used for feeding and/or loading newly emerged or field caught adult flies. In the center at the bottom of the front wall, a sliding door, 20 cm in width and 5 cm in height, was made to enable personnel to clean and to introduce oviposition medium in the cage.

**Maintenance of adults and egg collection.** Cages containing field caught *Stomoxys* flies were placed in the vivarium of the entomology section of the RIBSP and maintained at 24˚C and 85% humidity. Fresh citrated bovine blood soaked into cotton pads was provided twice daily at 9 a.m. and at 5 p.m.

Based on the protocol published by Parr [9] a modified medium was designed containing 400g dung and 200g clotted bovine blood. The manure mixed with hay was collected from the insect secured animal facility of RIBSP to eliminate any contamination by other insects. Blood was collected in glass bottles from the abattoir during slaughter and then allowed to clot for 30 min. These components were mixed (Viscosity stirrer AISI 316 steel, Bochem, Germany) and

incubated at room temperature overnight. The mixture obtained was equally divided into several shallow plastic pans 15 cm in diameter and 3 cm in height. The pans were then placed into the cages. Gravid female *Stomoxys* flies readily laid eggs in the crevices of the medium. Eggs were also found to a depth of 0.5 cm beneath the surface. The pans were removed twice a day and placed into the incubation chamber for further development.

### Effects of temperature and relative humidity on the development stages of the immature forms of *Stomoxys* species

Experiments were conducted to determine the ideal temperature and humidity range during the incubation to obtain a high degree of hatching. In addition to this, the effect of these factors on the viability of larva and pupae stages as well as on imago emergence was investigated. Gravid female flies were placed in the cylindrical vessels and allowed to oviposit. Eggs collected were divided into batches. Several batches each containing 150 eggs of each *Stomoxys* species were placed into circular pans 15 cm in diameter and 3 cm in depth. In each pan, 300 g medium was added and incubated separately using different incubation settings, namely at temperatures of 32˚C, 27˚C, 22˚C and relative humidity (RH) of either 90% (experiment 1) or 70% (experiment 2). The experiments were run in triplicates and mean values calculated.

The larval medium was placed into incubation chamber 6 hours prior to experiment commencement and the temperature of the substrate was measured hourly using Type K Thermocouples (Australia) to avoid any fluctuations in the incubation parameters. The medium was monitored every 4 hours for presence any changes in the development of immature *Stomoxys* flies as well as to calculate mean periods of incubation.

In preliminary studies, significant effects of some incubation conditions on the viability and yield of certain stages of immature *Stomoxys* flies were detected. This led to a consideration of utilizing various incubation parameters within a single life cycle of *Stomoxys* flies. To do so, experiment 3 was carried out using a combination of environmental settings, which had demonstrated potential significance on the development of immature stable flies. The combination method was performed as follows: egg maturation was conducted at 32˚C, 90% RH; incubation from larva until pupation completion at 27˚C, 70% RH; from pupae to imago eclosion at 27˚C, 70% RH. The content of the medium used in experiment 3 was similar to that utilized in experiments 1 and 2.

The mean time period required for completion of each developmental stage was not extended or reduced specifically to obtain a high yield of immature flies and was determined according to Friesen, Berkebile [10]. The experiment was run in triplicate and mean values were calculated.

### Data analysis

Data were analysed with logistic ANOVA and Tukey-Kramer multiple comparisons test in SAS 9.4 (Proc GLIMMIX, Cary, NC, USA). The dependent variables eggs hatched / total eggs, pupa / eggs hatched, and adults / pupae were assessed relative to the independent variables species, temperature, humidity and each of their 2-way interactions in separate analyses for each of the dependent variables. A value of $p \leq 0.05$ was designated statistically significant. Data demonstrating non-significant differences in their interactions were removed from the model in a stepwise manner

## Results

*Stomoxys sitiens* eggs hatched at a lower rate than the other two species (61% vs 68% for Sc and Si; Table 1). No interaction between species and either temperature or humidity was observed

**Table 1. Results of full model GLMM analyses.**

| Effect | df | % Hatch | | % Pupariation | | % Adult Emergence | |
|---|---|---|---|---|---|---|---|
| | | F | P | F | P | F | P |
| Species | 2,40 | 4.06 | 0.0249 | 0.03 | 0.9715 | 2.49 | 0.0955 |
| Temperature | 2,40 | 25.10 | <0.0001 | 19.25 | <0.0001 | 13.67 | <0.0001 |
| Humidity | 1,40 | 6.47 | 0.0149 | 0.02 | 0.8834 | 1.85 | 0.1810 |
| Species × Temp | 4,40 | 1.43 | 0.2423 | 1.21 | 0.3210 | 1.79 | 0.1500 |
| Species × Humidity | 2,40 | 2.60 | 0.0869 | 0.32 | 0.7277 | 0.47 | 0.6256 |
| Temp × Humidity | 2,40 | 21.13 | <0.0001 | 9.15 | 0.0005 | 3.50 | 0.0398 |

indicating the basic shapes of the relationships are similar. Temperature had a significant effect on hatch, however the relationship was curvilinear. Hatch was highest at 32˚C and lowest at 27˚C. Hatch was higher at 90% humidity (68.3% vs 62.8% at 70% humidity) and temperature and humidity had a significant interaction. Hatch rate was highest for all three species at 32˚ C and 90% humidity (Table 2).

No differences were observed in larval survival (hatched -> pupated) among species nor in the interactions between species and temperature or species and humidity. For this reason, species and species interactions were eliminated from the model. In the reduced model, temperature (F = 20.45, df = 2,48, P<0.0001) and the interaction between temperature and humidity (F = 9.59, df = 2,48, P = 0.0003) were both significant. Humidity had no effect on rate of pupariation (F = 0.02, df = 1,48, P = 0.8997). The relationship was curvilinear, the number of larvae successfully pupated was higher at 27˚C than either 22˚C (t = -4.89 df = 48, P<0.0001) or 32˚C (t = 6.17, df = 48, P<0.0001).

For adult emergence, the results were similar to those for larval survival. Neither species nor its interactions with temperature or humidity were significant. In the reduced model, emergence was highest at 27˚C (F = 13.01, df = 2,48, P<0.0001). Humidity had no effect on adult emergence (F = 1.34, df = 1,48, P = 0.2524).

## Combined method

Combined treatment demonstrated a higher egg to adult survival of *Stomoxys* spp than the constant condition treatments (46.2% vs 6.7–24.9 for the constant treatments; t = -11.26 - -6.32, df = 56, adjusted P<0.0001). The survival percentage for a total number of 150 eggs incubated under 32˚C, 90% RH was above 80% whereas the yield percentage for larvae and pupae maintained under 27˚C, 70% RH varied slightly averaging 55% and 48% respectively. The percentage of pupariation in the combined study did not differ from the same treatment in the constant condition study (*P* = 1.000). Adult emergence was higher than in the comparable treatment in the previous experiment (*P* ≤ 0.02). The mean quantitative indicators obtained from experiment 3 are shown in Table 3.

## Discussion

Blood feeding insects are vectors of infectious diseases of animals [13–16], including lumpy skin disease virus of cattle [8, 17, 18]. In addition, they can cause significant direct damage to livestock and discomfort and stress to companion animals and humans. Such insects include flies of *Stomoxys* species. To study the role of these flies inhabiting the territory of the Republic of Kazakhstan in the epizootiology of LSD, we have shown that three species of *Stomoxys* flies are capable of transmitting LSDV from diseased to healthy animals using a fractional feeding method [8]. In connection to this study, the work presented here aimed to study the biology of

**Table 2. Summary of incubation parameters affecting on immature forms of *Stomoxys* species.**

| Species | t° | Stage of development | | | | | | | | |
|---|---|---|---|---|---|---|---|---|---|---|
| | | Eggs* | | | Larvae** | | | Pupae*** | | |
| | | Period, hours/mean | Number of hatched/ mean±SD | % hatched | Period, hours/mean | Number of pupated/ mean±SD | % pupated | Period, hours/ mean | Number of imago emerged/ mean±SD | % emerged |
| | | Relative humidity 90% | | | | | | | | |
| SC | 32° | 25 (12–30) | 119±8.50 | 79.3 | 240 (235–250) | 49±9.01 | 33 | 144 (140–150) | 22±6.80 | 17 |
| SS | | 28 (13–32) | 126±9.45 | 82 | 240 (220–255) | 32±6.55 | 21 | 120 (96–144) | 12±5.03 | 8 |
| SI | | 22 (15–27) | 133±6.02 | 89 | 264 (245–276) | 51± 12.8 | 34 | 120 (100–138) | 14±4.72 | 9 |
| SC | 27° | 27 (12–31) | 96±13.2 | 64 | 216 (204–228) | 57±14.0 | 38 | 48 (40–72) | 32±11.0 | 21 |
| SS | | 24 (18–29) | 72±11.9 | 48 | 216 (200–228) | 44±9.60 | 29 | 72 (50–84) | 30±9.60 | 20 |
| SI | | 24 (10–29) | 87±9.71 | 58 | 240 (216–260) | 60±12.2 | 40 | 48 (40–56) | 29±9.60 | 19 |
| SC | 2° | 30 (18–35) | 99±8.50 | 66 | 312 (264–327) | 59±3.51 | 39 | 52 (45–60) | 24±7.55 | 16 |
| SS | | 26 (19–32) | 71±9.01 | 60,6 | 288 (238–306) | 44±7.02 | 29 | 72 (60–85) | 30±8.02 | 20 |
| SI | | 27 (16–33) | 94±9.01 | 47 | 288 (230–312) | 47±6.50 | 31 | 72 (60–84) | 27±9.53 | 18 |
| | | Relative humidity 70% | | | | | | | | |
| SC | 32° | 30 (21–33) | 102±10.5 | 68 | 240 (216–264) | 44±10.6 | 29 | 96 (90–102) | 18±7.55 | 12 |
| SS | | 26 (19–29) | 89±12.7 | 59 | 240 (200–252) | 39±4.61 | 26 | 96 (86–106) | 22±6.55 | 14 |
| SI | | 25 (16–30) | 93±15.7 | 62 | 240 (228–250) | 51±9.61 | 34 | 96 (90–102) | 11±3.21 | 7.3 |
| SC | 27° | 24 (15–29) | 84±7.67 | 56 | 240 (210–256) | 62±11.5 | 41 | 72 (65–80) | 44±11.1 | 29 |
| SS | | 24 (17–26) | 93±10.1 | 62 | 240 (220–246) | 69±14.2 | 46 | 96 (84–108) | 39±7.51 | 26 |
| SI | | 24 (15–28) | 75±9.45 | 50 | 216 (202–228) | 51±9.01 | 34 | 72 (66–78) | 42±11.3 | 28 |
| SC | 22° | 26 (16–29) | 101±9.07 | 67 | 288 (240–312) | 34±7.02 | 22 | 72 (67–96) | 9±3.05 | 6 |
| SS | | 34 (20–38) | 96±5.56 | 64 | 288 (246–300) | 40±9.53 | 26 | 72 (62–96) | 13±4.51 | 8.6 |
| SI | | 31 (17–37) | 112±11.1 | 75 | 264 (236–294) | 36±10.6 | 24 | 72 (65–96) | 7±4.04 | 4,6 |

Digits in parentheses represents the mean onset and completion time of hatching, pupation and eclosion periods in triplicates.

(*)–A total number of eggs used in the experiment was 150eggs for each *Stomoxys* species.

(**)—The number of larva pupated from first instar hatched.

(***)The number of imago emerged from viable pupae.

SC—*Stomoxys calcitrans*.

SS—*Stomoxys sitiens*.

SI—*Stomoxys indica*.

reproduction of these flies and, based on the results of which, modelling the microclimate for their maintenance and reproduction under laboratory conditions. The created model will

**Table 3. Quantitative indicators of the development of immature flies using combined incubation parameters in a single life cycle.**

| Species | Stage of development | | | | | | | | |
|---|---|---|---|---|---|---|---|---|---|
| | Eggs (32°, 90% RH) | | | Larvae (27°, 70% RH) | | | Pupae (27°, 70% RH) | | |
| | Period, hours/mean | Number of hatched/mean | % hatched | Period, hours/mean | Number of pupated/mean | % pupated | Period, hours/mean | Number of imago emerged/mean | % emerged |
| SC | 29 (15–33) | 115±15.04 | 76.6 | 240 (204–268) | 84±9.53 | 56 | 144 (120–156) | 62±10.7 | 41 |
| SS | 27 (17–30) | 131±9.45 | 87.3 | 240 (220–262) | 97±11.01 | 49 | 132 (108–156) | 77±8.73 | 51 |
| SI | 23 (15–28) | 120±7.02 | 80 | 240 (204–280) | 87±8.62 | 58 | 120 (108–140) | 70±9.01 | 53 |

SC—*Stomoxys calcitrans*.

SS—*Stomoxys sitiens*.

SI—*Stomoxys indica*.

subsequently be used in studies evaluating target flies in the role of the mechanical and biological vector of the causative agent of LSD and other vector-borne diseases during spread within the herd and reservation in the natural niche in the interepizootic period.

As the study results show, the developmental biology of all three species of *Stomoxys* flies was found to be similar. Immature forms of all three species (SC, SS, SI) at the stages of eggs, larva, pupa and imago under the same microclimate conditions developed satisfactorily and similarly. In this study, most of the results obtained align with previous results on the effects of ambient environment on the species' survival and developmental time under laboratory condition [19, 20]. As for the tested microclimate parameters, we selected two levels of relative humidity, which were maintained at 70% or 90%, and the ambient temperature was selected as equal to 22°C, 27°C, or 32°C. The selection of these parameters was based on studies reported by Kunz, Berry [19], Wang and Gili [21] and Ramsamy [22] Observation of eggs laid and maintained under the indicated microclimate conditions showed that the duration of the hatching period at various temperatures and different humidity were approximately similar to each other, except for a temperature of 22°C, at which this time was extended by 2.6 hours at 90% humidity and by 3.3–6.3 hours at 70% humidity compared with other temperature settings. This observation is in agreement with findings reported by Lysyk [23].

At both humidity indices, the predominantly favorable parameter for the formation of pupae was observed when incubated at 27°C, at which the maximum number of pupae were obtained (35.6% of pupae at 90% humidity and 40.3% of pupae at 70% humidity). The results obtained demonstrate that pupation most favorably occurs at incubation under 27°C and 70% relative humidity. This, in turn, is consistent with previous studies where larvae selected moderate moisture levels in which to pupate avoiding excessively high and low moisture levels [24, 25].

The data obtained during the observation of the final biological development stage of flies demonstrates that humidity and ambient temperature play a decisive role in the active formation and emerging of adult insects. The most suitable indicators of these parameters were humidity equal to 70% and temperature of 27°C.

Analysis of the study results indicates that for the active development of flies and their reproduction in high numbers, a certain microclimate is required [26–28]. At the stage of eggs, an increased humidity of 90% and a relatively high temperature reaching 32°C is favorable and productive for their transition into a larval stage. In this mode, up to 80% of larvae can be obtained from the initial number of eggs laid by an adult fly. Probably, the high humidity level is essential for the formation of larvae. Therefore, they burrow deeper into the medium, where the humidity is higher, from the level at which the eggs were laid.

At the next stage of biological development, when pupation occurs, the demand for higher humidity decreases. A possible explanation for this could be a lower moisture requirement to form a puparium as well as the number of formed pupae becomes relatively small. Therefore, a decrease in humidity at this stage of development stimulates the formation of an increased number of pupae, particularly when incubated at a temperature of 27°C. The benefit of this temperature for the formation of pupae is noted both at high (90%) and low (70%) humidity. Therefore at optimal temperature, humidity is not important (within the range tested). However, at low temperatures, humidity has a significant effect. At high temperature, humidity affected larval survival, but not adult emergence. Effect of temperature on pupariation (22°C, 32°C) did not differ and appeared to be curvilinear. On contrary, pupa formation under 27°C was significantly higher than either $P < 0,0001$. Temperature and temperature—humidity interaction (marginal) have significant effects on adult emergence $P < 0,0001$.

Temperature limits above (32°C) or below (22°C) were not evaluated in this study. This was due to an earlier study which showed that constant incubation temperatures at 35°C, 20°C

and 15˚C resulted in high mortalities in immature *Stomoxys* forms [20]. For *Stomoxys calcitran*s, egg to adult survival for all of the constant temperatures was similar to those observed by Florez-Cuadros, Berkebile [29]. Maintaining an incubation microclimate with a humidity of 70% and a temperature of 27˚ C provided a large population of *Stomoxys* spp under laboratory settings, reaching 70% or more of the number of stock pupae available.

In addition, the larval medium used in this study has proven suitable for rearing *Stomoxys* immature forms. Initially, to induce egg laying, a medium was prepared according to Parr (1959) (300g dried cow dung, 150g dried blood granules and 30g sugar). Eggs introduced into this medium demonstrated a meagre percentage of hatching, and those that did hatch did not survive the first instar stage. On reflection, it was hypothesized that dried dung and dried blood became sterile during the medium preparation procedure where ingredients were subjected to complete drying. This is in agreement with findings reported in other studies [30–33] indicating that bacteria present in the natural substrate sustain the viability of stable flies throughout developmental stages. Moreover, production cost is an essential consideration in any stable fly maintenance program. A lack of some costly ingredients in the larval medium used such as a wheat barn, sugar cane bagasse, fishmeal and wood shavings and ammonium bicarbonate may have an effect on the viability of immature *Stomoxys* flies however this was not evaluated in this study.

## Conclusion

To study the biology of the reproduction of *Stomoxys* flies and the selection of the optimal microclimate for their breeding, seven microclimates were designed and tested, with different temperature conditions and air humidity. Using incubation in double-level microclimate have proven to be efficient at the level of production we have been maintaining.

## Supporting information

**S1 File. Comparison of egg to adult survival of combined regiment to that on the constant regiments.**
(ZIP)

**S2 File.**
(PDF)

## Acknowledgments

The author is sincerely thankful to Mr Aslan Kerembaev and Ms Raihan Nissanova for technical assistance provided.

## Author Contributions

**Conceptualization:** Kuandyk Zhugunissov, Aliya Akhmetaliyeva, Malik Shalmenov, Peter J. White.

**Data curation:** Kuandyk Zhugunissov, Nurlybay Kazhgaliyev, Malik Shalmenov.

**Formal analysis:** Malik Shalmenov, Gaisa Absatirov.

**Funding acquisition:** Gaisa Absatirov, Peter J. White.

**Investigation:** Lespek Kutumbetov, Nurlybay Kazhgaliyev, Aliya Akhmetaliyeva, Gaisa Absatirov.

**Methodology:** Lespek Kutumbetov, Nurlybay Kazhgaliyev, Aliya Akhmetaliyeva, Laura Dushayeva.

**Project administration:** Arman Issimov.

**Resources:** Laura Dushayeva.

**Software:** David B. Taylor, Kuandyk Zhugunissov, Assylbek Zhanabayev, Laura Dushayeva.

**Supervision:** Arman Issimov, David B. Taylor, Assylbek Zhanabayev, Peter J. White.

**Validation:** Assylbek Zhanabayev.

**Visualization:** Aliya Akhmetaliyeva, Birzhan Nurgaliyev.

**Writing – original draft:** Arman Issimov.

**Writing – review & editing:** David B. Taylor, Birzhan Nurgaliyev, Peter J. White.

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
