## [Decision Letter · Decision Letter 0]

27 May 2020

PONE-D-20-12464

The combined effects of temperature and relative humidity parameters on the reproduction of Stomoxys species in a laboratory setting

PLOS ONE

Dear Dr. Issimov,

Thank you for submitting your manuscript to PLOS ONE. After careful consideration, we feel that it has merit but does not fully meet PLOS ONE’s publication criteria as it currently stands. Therefore, we invite you to submit a revised version of the manuscript that addresses the points raised during the review process.

We look forward to receiving your revised manuscript.

Kind regards,

Olle Terenius

Academic Editor

PLOS ONE

Journal Requirements:

2. In your Methods section, please provide additional information regarding the permits you obtained for the work. Please ensure you have included the full name of the authority that approved the collection sites access and, if no permits were required, a brief statement explaining why.

Reviewers' comments:

Reviewer's Responses to Questions

**Comments to the Author**

1. Is the manuscript technically sound, and do the data support the conclusions?

Reviewer #1: Yes

2. Has the statistical analysis been performed appropriately and rigorously? 

Reviewer #1: No

3. Have the authors made all data underlying the findings in their manuscript fully available?

Reviewer #1: No

4. Is the manuscript presented in an intelligible fashion and written in standard English?

Reviewer #1: Yes

5. Review Comments to the Author

Reviewer #1: Issimov et al The combined effects of temperature and relative humidity parameters on the reproduction of Stomoxys species in a laboratory setting

This manuscript presents a series of experiments to determine the optimal temperature and humidity levels for rearing three species of Stomoxys. The information is important, especially for S. sitiens and S. indica, two species for which limited information is available. The experimental design appears to be adequate for answering the questions asked. However, some clarification of methods is needed. The statistical analysis needs clarification and is probably inadequate. This renders the results section difficult to understand. The last paragraph of the discussion extends well beyond the experiments and data provided in the manuscript. Resolution of the figures is poor. I have included a copy of the manuscript with some suggestions for improving grammar, readability, and specific questions/comments. The manuscript requires major revision before it can be reconsidered for publication.

Statistical analysis.

The model for the statistical analysis is not clear. Given my understanding of the experimental design, the model should be something along the lines of x=Humidity × Temperature × Species × interactions. The manuscript indicates a 2-way ANOVA, presumably with the model x=Humidity × Temperature. It is not clear if the interaction term was evaluated or not. The results are presented as if each treatment (Humidity × Temperature) was independent. The results should first report if there was an effect for each of the primary independent variables, humidity, temperature and species. Next the interactions should be addressed. As currently written, the similarity of species within each treatment is discussed, but I do not see any statistical support. A proper model followed by reorganization of the Results section to follow the model will improve the manuscript greatly. The data being analyzed are count data. Either logistic or Poisson/Negative Binomial models would be more appropriate. The authors to not discuss the fit of their data to the model used.

The last paragraph of the discussion section indicates the presented methods are superior to those previously published. However, no data on experiments employing the previously published methods are provided. Data are required to make this comparison. Overall, egg to adult survival is quite low for all of the constant temperature and humidity treatments (compare to Florez-Cuadros et al 2019) which used a similar experimental design. I am actually surprised the authors were able to get stable fly larvae to survive on the substrate described in this paper. It appears to be way too dense and rich for stable flies. Natural stable fly substrates will always have some vegetative material. Despite years of sampling, I have never seen stable fly larvae developing in pure, fresh bovine feces. If the authors are claiming this substrate, without vegetative material, is superior, I will need to see some data. Some indication of size of the flies reared on their diet would be helpful as well. It is one thing to have good survival, it is another to have flies comparable in size to those collected in the wild.

6. PLOS authors have the option to publish the peer review history of their article (what does this mean?). If published, this will include your full peer review and any attached files.

Reviewer #1: Yes: David B Taylor

---

## [Author Response · Author response to Decision Letter 0]

9 Oct 2020

Given manuscript is written according to PLOS ONE's requirements. 

Additional information regarding to permit for work were added to Method section.

---

## [Editor Report · Decision Letter 1]

10 Nov 2020

The combined effects of temperature and relative humidity parameters on the reproduction of Stomoxys species in a laboratory setting

PONE-D-20-12464R1

Dear Dr. Issimov,

We’re pleased to inform you that your manuscript has been judged scientifically suitable for publication and will be formally accepted for publication once it meets all outstanding technical requirements.

Kind regards,

Olle Terenius

Academic Editor

PLOS ONE
---

## [Editor Report · Acceptance letter]

26 Nov 2020

PONE-D-20-12464R1 

The combined effects of temperature and relative humidity parameters on the reproduction of Stomoxys species in a laboratory setting 

Dear Dr. Issimov:

I'm pleased to inform you that your manuscript has been deemed suitable for publication in PLOS ONE. Congratulations! Your manuscript is now with our production department. 

Kind regards, 

on behalf of

Dr. Olle Terenius 

Academic Editor

PLOS ONE